# Advances in the Fabrication, Properties, and Applications of Electrospun PEDOT-Based Conductive Nanofibers

**DOI:** 10.3390/polym16172514

**Published:** 2024-09-04

**Authors:** Emanuele Alberto Slejko, Giovanni Carraro, Xiongchuan Huang, Marco Smerieri

**Affiliations:** 1IMEM-CNR, Institute of Materials for Electronics and Magnetism of the National Research Council of Italy, Via Dodecaneso 33, 16146 Genova, Italy; giovanni.carraro@cnr.it (G.C.); marco.smerieri@cnr.it (M.S.); 2School of Information Science and Technology, Fudan University, Handan Rd. 220, Shanghai 200433, China; huangxiongchuan@fudan.edu.cn

**Keywords:** nonwovens, conductive polymer, electrospinning, PEDOT:PSS, electrospun nanofibers

## Abstract

The production of nanofibers has become a significant area of research due to their unique properties and diverse applications in various fields, such as biomedicine, textiles, energy, and environmental science. Electrospinning, a versatile and scalable technique, has gained considerable attention for its ability to fabricate nanofibers with tailored properties. Among the wide array of conductive polymers, poly(3,4-ethylenedioxythiophene) (PEDOT) has emerged as a promising material due to its exceptional conductivity, environmental stability, and ease of synthesis. The electrospinning of PEDOT-based nanofibers offers tunable electrical and optical properties, making them suitable for applications in organic electronics, energy storage, biomedicine, and wearable technology. This review, with its comprehensive exploration of the fabrication, properties, and applications of PEDOT nanofibers produced via electrospinning, provides a wealth of knowledge and insights into leveraging the full potential of PEDOT nanofibers in next-generation electronic and functional devices by examining recent advancements in the synthesis, functionalization, and post-treatment methods of PEDOT nanofibers. Furthermore, the review identifies current challenges, future directions, and potential strategies to address scalability, reproducibility, stability, and integration into practical devices, offering a comprehensive resource on conductive nanofibers.

## 1. Introduction

The production of nanofibers has emerged as a pivotal area of research due to their unique properties and wide-ranging applications in fields such as biomedicine, textiles, energy, and environmental science [1,2]. Among the various methods available for nanofiber fabrication, electrospinning has garnered significant attention for its versatility, scalability, and ability to produce nanofibers with tailored properties [3]. Electrospinning is a technique that involves applying a high-voltage electrical field to a polymer solution or melt. This process induces the formation of a thin jet, which elongates and solidifies into continuous nanofibers as the solvent evaporates or the polymer solidifies. The resulting nanofibers exhibit diameters ranging from a few nanometers to several micrometers, with exceptional uniformity and purity.

One advantage of electrospun nanofibers is their high surface area-to-volume ratio, which enhances their interactions with surrounding environments [4]. This property makes electrospun nanofibers suitable for filtration, tissue engineering, drug delivery, and sensing applications. Moreover, the properties of electrospun nanofibers can be easily tailored by adjusting various parameters, including polymer concentration, solvent type, applied voltage, and collection distance, among many others [5]. This tunability enables the customization of nanofibers to meet specific application requirements, further enhancing their utility across diverse fields. In addition to their tunable properties, electrospun nanofibers offer scalability and cost-effectiveness, making them suitable for industrial production. The simplicity of the electrospinning process, coupled with its ability to produce uniform and pure nanofibers, makes it an attractive option for large-scale manufacturing.

Conductive polymers have garnered significant attention in recent years for their unique combination of electrical conductivity, mechanical flexibility, and processability, opening new avenues for developing advanced electronic devices and energy storage systems [6]. Several experiments have been conducted on polyaniline (PANI), polypyrrole (PPy), polythiophenes, and many others [7]. Among the various conductive polymers, poly(3,4-ethylenedioxythiophene) (PEDOT) is a promising material due to its exceptional conductivity, environmental stability, and ease of synthesis. PEDOT belongs to the family of polythiophenes and is characterized by its π-conjugated structure, consisting of alternating double and single bonds along the polymer backbone [5]. This π-conjugation imparts high electrical conductivity to PEDOT, making it one of the most conductive polymers known to date. Furthermore, ethylenedioxy (EDOT) functional groups enhance PEDOT’s solubility and processability, enabling facile fabrication of thin films, coatings, and composites. The first scientific contribution investigating the fabrication and properties of PEDOT-based nanofibers is dated 2008 [8]. The highest PEDOT conductivity reported, approximately 4380 S/cm and almost comparable to that of ITO, was achieved through a controlled sulfuric acid post-treatment, which induced the formation of crystallized PEDOT nanofibrils by removing PSS via charge separation [9].

One of the key advantages of PEDOT is its environmental stability, attributed to the presence of ether linkages in the polymer backbone. PEDOT exhibits remarkable stability under ambient conditions, unlike many other conductive polymers, which are prone to degradation upon air, moisture, or UV radiation exposure. This property makes PEDOT particularly well suited for applications requiring long-term reliability and durability, such as organic electronic devices and biosensors. Moreover, PEDOT offers tunable electrical and optical properties through chemical doping and structural modifications. By doping PEDOT with various dopant molecules, such as polyanions or conducting salts, its conductivity can be tailored over a wide range to meet specific application requirements. Additionally, the optical transparency of PEDOT can be adjusted by controlling the polymer chain morphology, making it suitable for optoelectronic applications, including transparent electrodes and displays. In recent years, PEDOT has found widespread use in diverse fields, including organic electronics, energy storage, biomedicine, and wearable technology [1,2]. Its compatibility with flexible substrates and biocompatibility has made PEDOT an attractive candidate for flexible electronics, implantable medical devices, and bioelectronic interfaces [10,11]. Furthermore, its high charge storage capacity and rapid redox kinetics have enabled the development of high-performance supercapacitors and batteries with enhanced cycling stability and power density.

The most common commercially available product of PEDOT is its dispersion in water. PEDOT is stabilized by polystyrene sulfonate (PSS), which balances the positively charged PEDOT thanks to its negative charges (Figure 1). Several investigations have been conducted on the morphology and chemical configuration of PEDOT:PSS water dispersions: studies have validated the prevailing model of PEDOT films [12], confirming that they are composed of spherical grains with diameters ranging from 50 to 80 nm. Energy-dispersive X-ray analysis indicates that these grains likely have a 5–10 nm thick PSS-rich shell surrounding a PEDOT-rich core. High-magnification High-Angle Annular Dark-Field Scanning Transmission Electron Microscopy images suggest that the grains are made up of individual tangles approximately 5 nm or smaller in diameter. The cohesion between grains is primarily achieved through hydrogen bonds between PSS groups in the shells. In the context of PEDOT:PSS, electrospinning offers several advantages. First, it enables the fabrication of nanofibers with high surface area-to-volume ratios [4], which is beneficial for applications such as sensors, actuators, and tissue engineering [13]. Second, it allows for precise control over fiber diameter and alignment, which can influence properties like conductivity and mechanical strength. Third, electrospinning can be used to create complex three-dimensional structures, providing versatility in design. The significance of electrospinning for PEDOT:PSS nanofibers lies in its potential to enhance the material’s performance and expand its applications in bioelectronics. By producing PEDOT:PSS nanofibers through electrospinning, researchers can create bioactive interfaces with enhanced conductivity, mechanical properties, and biocompatibility. These interfaces are crucial for developing advanced biomedical devices such as biosensors, neural implants, and drug delivery systems. The evolution of PEDOT nanofibers produced by electrospinning has seen significant advancements, particularly in their conductivity and mechanical properties. Initially, conductive PEDOT nanofibers were fabricated through electrospinning of insulating polymers followed by vapor-phase polymerization of PEDOT, achieving high conductivity and resistive heating capabilities [14,15]. It was only during the 2010s that studies introduced techniques like oxidative polymerization and the blending of PEDOT with other polymers for solution-based electrospinning, further enhancing the structural stability, conductivity, and flexibility of the nanofibers [16,17,18]. In 2017, Yu et al. were able to propose a one-step fabrication of core–shell nanofibers based on PEDOT:PSS and poly(ethylene oxide), while three years later, Du et al. developed a core–shell structure with chitosan nanofibers with improved biocompatibility for biomedical research and regenerative medicine [19,20]. These innovations have positioned PEDOT nanofibers as promising materials for applications in flexible electronics, sensors, and biomedical devices.

This review systematically examines the state-of-the-art techniques for producing PEDOT nanofibers via electrospinning, enhancing their electrical conductivity, and exploring their diverse applications. Firstly, we provide an overview of the electrospinning process, discussing key parameters such as solution properties, processing conditions, and collector configuration that influence the morphology and properties of the resulting nanofibers. Next, we delve into recent advancements in the synthesis and functionalization of PEDOT-based precursors for electrospinning, including the incorporation of dopants, additives, and nanomaterials to tailor the electrical conductivity and mechanical properties of the nanofibers. Furthermore, we explore various post-treatment methods employed to enhance further the conductivity of PEDOT nanofibers, such as chemical doping, thermal annealing, and surface modification, evaluating their effectiveness in improving electrical performance while preserving nanofiber morphology. Subsequently, we investigate the emerging applications of PEDOT nanofibers in electronic devices, energy storage systems, sensors, actuators, and biomedical devices, highlighting their unique advantages, such as high surface-to-volume ratio, flexibility, and biocompatibility [2]. By consolidating recent research findings and insights from the literature, this work provides researchers and engineers with a comprehensive understanding of the opportunities and challenges associated with PEDOT nanofiber production via electrospinning. Finally, we identify current challenges and future directions in the field, including scalability, reproducibility, stability, and integration into practical devices, and propose potential strategies to address these issues. Overall, it serves as a valuable resource for researchers seeking to harness the full potential of PEDOT nanofibers in next-generation electronic and functional devices.

## 2. Fundamentals of Electrospinning

Electrospinning is a versatile and widely used technique for producing nanofibers with controllable morphology, high surface area, and diverse properties. The process involves the application of an electric field to a polymer solution or melt, causing the formation of an electrified jet that undergoes stretching and solidification to form nanofibers. The physical mechanisms underlying electrospinning are complex and involve several key phenomena, including electrostatic forces, surface tension, fluid dynamics, and polymer chain entanglement. Understanding these mechanisms is crucial for optimizing the electrospinning process and tailoring the properties of the resulting nanofibers.

At the heart of electrospinning is the interaction between electric fields and polymer solutions or melts [21,22]. When a high electric field is applied to a polymer solution or melt, charges accumulate on the surface of the droplet or meniscus due to the electrostatic induction. The repulsion between like charges leads to the formation of a Taylor cone at the tip of the droplet or meniscus, where the electric field strength is the highest. The curvature of the Taylor cone increases the electric field strength, promoting the ejection of a fine jet of polymer solution or melt from the cone apex. The stretching and solidification of the ejected jet are governed by a balance of forces, including electrostatic forces, surface tension, and viscous forces. As the jet travels towards the collector under the influence of the electric field, it undergoes elongation and thinning due to the electrostatic repulsion between charged polymer chains. The stretching of the jet promotes molecular alignment and orientation along the axis of the fiber, resulting in nanofibers with high aspect ratios and aligned structures. Surface tension also plays a crucial role in electrospinning by resisting the elongation of the jet and promoting the formation of a stable Taylor cone. The surface tension of the polymer solution or melt determines the size and shape of the Taylor cone, as well as the stability of the jet during flight. Low-surface-tension solutions tend to form finer and more uniform nanofibers due to reduced instability and jet whipping. Fluid dynamics govern the motion of the polymer solution or melt during electrospinning and influence the morphology of the resulting nanofibers. The viscosity and rheological properties of the polymer solution or melt dictate the jet behavior, including its stretching, bending, and whipping. High-viscosity solutions tend to produce thicker and less uniform nanofibers due to increased resistance to stretching and bending. Polymer chain entanglement also plays a critical role in electrospinning by promoting the formation of stable jets and preventing premature jet breakup. Entangled polymer chains exhibit viscoelastic behavior, allowing them to deform and stretch under the influence of the electric field without undergoing catastrophic rupture. Polymer solutions with higher molecular weights and concentrations typically exhibit greater chain entanglement, forming finer and more uniform nanofibers.

A conventional solution-electrospinning setup, as illustrated in Figure 2, comprises several essential components, including a high-voltage power source, a syringe pump, a syringe loaded with the polymeric solution, a metal spinneret, usually a blunt-cut needle, connected to the syringe, and a fiber collector, typically a metal plate connected to a grounded electrode. The syringe pump regulates the flow rate of the polymer solution. The high-voltage power supply, in conjunction with the spinneret that directs the charged polymer solution into a high electric field and ultimately on the collector, provides the force necessary to elongate the charged polymer solution into fibrous structures.

Understanding the equipment setup and operational parameters is crucial for controlling the electrospinning process and tailoring the properties of the resulting nanofibers. The most important components are detailed in the following.

High-Voltage Power Supply: The high-voltage power supply is a critical component of the electrospinning setup, providing the electric field necessary to generate the electrospun jet. Voltages typically range from a few kilovolts to tens of kilovolts, depending on the specific polymer solution and desired fiber properties. The power supply must deliver a stable and controllable voltage to ensure consistent electrospinning performance.Syringe Pump or Spinneret: The polymer solution or molten polymer is commonly inserted into a syringe attached to either a syringe pump or spinneret. The syringe pump regulates the polymer solution flow, enabling accurate management of the deposition rate and fiber diameter. In contrast, a spinneret can extrude the polymer solution or molten polymer at a controlled rate.Collector: The collector is a conductive substrate placed at a fixed distance from the spinneret to collect the electrospun fibers. Depending on the desired fiber alignment and morphology, the collector can take various forms, including flat plates, rotating drums, or mandrels. The collector can also be grounded to provide an electrical path for the charged fibers, preventing charge accumulation and promoting uniform deposition.Grounded Electrode: A grounded electrode is typically placed close to the collector to provide an electrical path for the charged polymer solution or melt. The grounded electrode helps to neutralize the charges on the collected fibers, preventing them from repelling each other and forming aggregates. The grounded electrode can be positioned below or surrounding the collector, depending on the specific electrospinning setup.

Operational parameters for the electrospinning process must be set up depending on the fabrication of single polymeric nanofibers. The most important ones include the following:Flow Rate: The flow rate of the polymer solution or melt is a critical parameter that determines the deposition rate and fiber diameter during electrospinning. Higher flow rates generally result in thicker fibers due to increased polymer deposition, while lower flow rates produce finer fibers. The flow rate can be adjusted using the syringe pump or spinneret to optimize the electrospinning process for a given polymer system.Voltage: The applied voltage plays a crucial role in controlling the electrospinning process by governing the formation and behavior of the electrified jet. Higher voltages increase electrostatic repulsion between the charged polymer chains, promoting jet elongation and thinning. However, excessively high voltages can cause jet instability and lead to beads or non-uniform fibers forming. The voltage should be optimized based on the polymer solution properties and desired fiber morphology.Distance: The distance between the spinneret and the collector is another important parameter influencing the electrospinning process. The distance affects the trajectory and stretching of the ejected jet and the deposition pattern and fiber alignment on the collector. Closer distances generally result in finer fibers due to increased stretching, while larger distances produce thicker fibers. The distance should be carefully adjusted to achieve the desired fiber morphology and alignment.Solution Properties: The properties of the polymer solution or melt, including viscosity, conductivity, surface tension, and concentration, influence the electrospinning process and fiber morphology. Higher-viscosity solutions tend to produce thicker fibers due to increased resistance to jet elongation, while lower-viscosity solutions produce finer fibers. Similarly, higher-conductivity solutions promote more efficient charge dissipation and jet stabilization, leading to finer and more uniform fibers. The concentration of the polymer solution also affects the fiber morphology, with higher concentrations generally resulting in thicker fibers.

## 3. Synthesis of PEDOT-Based Nanofibers via Electrospinning

Electrospinning is a highly effective technique for fabricating nanofibers, offering advantages such as simplicity, versatility, and cost-effectiveness. PEDOT stands out among the various materials suitable for electrospinning due to its excellent electrical conductivity, environmental stability, and biocompatibility [10]. However, as PEDOT is a conductive material, other non-conducting polymers or chemicals, like poly(ethylene oxide)—PEO—are typically added to help with fiber production [5,24].

Electrospinning offers a unique advantage for fabricating PEDOT-based conductive nanofibers compared to other fabrication methods and fiber materials, particularly in terms of applications (see Table 1). This technique allows for the creation of nanofibers with high surface area, porosity, and flexibility, which are crucial for various advanced applications such as wearable electronics, energy storage, and biomedical devices. The ability to control fiber morphology and composition through electrospinning enhances the performance of PEDOT-based nanofibers, making them superior to other methods and materials. The specific advantages of electrospinning for PEDOT-based conductive nanofibers are the following.

High surface area and porosity: Electrospinning produces nanofibers with a high surface area-to-volume ratio, which is valuable for applications requiring high reactivity and interaction with the environment, such as sensors and catalysts [25].Flexibility and comfort: The nanofibers mats produced are highly flexible, making them suitable for wearable electronics where comfort and adaptability to body movements are essential [26].Enhanced conductivity: By incorporating materials like reduced graphene oxide (rGO), the conductivity of PEDOT-based nanofibers can be significantly enhanced, which is crucial for electronic applications [27].Versatility in material composition: Electrospinning allows for the incorporation of various materials, such as nanocarbons, to enhance the mechanical and electrical properties of the fibers, making them suitable for a wide range of applications from photovoltaics to biomedical devices [28].Improved mechanical properties: The process can improve the mechanical properties of the fibers, such as tensile strength, which is important for structural applications [27].Controlled morphology: Electrospinning allows for precise control over fiber diameter and morphology, which can be tailored to specific application needs, such as creating coaxial structures for electronic devices [29,30].

PEDOT nanofibers have been synthesized using several electrospinning methods, including solution-based and melt-electrospinning. Each method has distinct advantages and limitations, which will be discussed in the following sections.

### 3.1. Solution-Based Electrospinning

Solution-based electrospinning is the most employed method for synthesizing PEDOT nanofibers. In this approach, PEDOT or its precursors are dissolved in a suitable solvent to form a homogeneous solution that can be electrospun into nanofibers. The first step in solution-based electrospinning is to prepare a polymer solution with appropriate viscosity, conductivity, and surface tension. For PEDOT nanofibers, PEDOT:PSS (poly(3,4-ethylenedioxythiophene):poly(styrenesulfonate)) is often used due to its solubility in water and various organic solvents. PEDOT:PSS solutions can be mixed with co-solvents such as dimethyl sulfoxide (DMSO) or ethylene glycol to improve the conductivity and film-forming properties of the resulting nanofibers. The electrospinning process begins by loading the polymer solution into a syringe with a needle or spinneret. A high-voltage power supply is connected to the needle, creating an electric field between the needle and a grounded collector. As the voltage increases, the electrostatic forces overcome the surface tension of the solution, forming a Taylor cone at the tip of the needle. A jet of polymer solution is then ejected from the Taylor cone and undergoes stretching and thinning as it travels toward the collector. Operational parameters such as flow rate, applied voltage, and the distance between the needle and collector (working distance) significantly influence the morphology and properties of the resulting nanofibers. For instance, a higher voltage can increase the stretching of the jet, leading to finer fibers, while the flow rate must be optimized to ensure a continuous and stable jet without bead formation. The electrified jet solidifies upon solvent evaporation, and the fibers are collected on a grounded substrate, which can be a flat plate, rotating drum, or wire mesh. The choice of collector affects the alignment and orientation of the nanofibers. For example, a rotating drum collector can produce aligned nanofibers, which is desirable for applications requiring anisotropic properties.

Solution-based electrospinning is straightforward and adaptable to a wide range of polymers, including PEDOT:PSS [31]. It allows for easy incorporation of various solvents and additives to modify the properties of the nanofibers. By adjusting the concentration of the polymer solution, applied voltage, flow rate, and working distance, the morphology of the nanofibers can be precisely controlled. This flexibility is beneficial for tailoring the fibers for specific applications. The equipment and materials required for solution-based electrospinning are relatively inexpensive and widely available, making it a cost-effective method for producing nanofibers. The main limitations of solution-based electrospinning are related to using solvents, which can introduce several challenges, including toxicity, volatility, and environmental concerns. Handling and disposing of solvents safely can also add to the overall cost and complexity of the process. Traces of solvent may remain in the nanofibers after electrospinning, potentially affecting their properties and performance, particularly in applications where purity is critical. Not all polymers are soluble in suitable solvents, which can limit the choice of materials that can be electrospun using this method.

One of the first routes to produce PEDOT-based nanofibers using solution-electrospinning relied on fabricating a carrier polymer nonwoven structure followed by the vapor deposition of PEDOT on the samples in a controlled environment [15]. This way, a PEDOT coating was obtained over the carrier polymer. More recently, the process has been optimized to electrospin PEDOT-based nanofibers directly from the starting solution without any subsequent vapor deposition [32].

### 3.2. Melt-Electrospinning

Melt-electrospinning, an advanced method for producing PEDOT nanofibers without solvents, requires high precision and control. This technique involves melting the polymer and electrospinning it in the molten state. Melt-electrospinning offers several advantages, including avoiding solvent-related issues such as toxicity, volatility, and environmental impact. However, it also demands higher temperatures and specialized equipment to handle the molten polymer. The polymer is heated above its melting point to obtain a molten state suitable for electrospinning, requiring precise temperature control to ensure a stable and homogeneous melt. Additives or plasticizers may be incorporated to lower the melting temperature and improve the spinnability of the polymer. Like solution-based electrospinning, melt-electrospinning involves applying a high voltage to the molten polymer, creating an electrified jet. However, due to the higher viscosity of the molten polymer compared to a solution, the applied voltage and working distance must be carefully controlled to achieve continuous and uniform fiber formation. Using a heated spinneret and a controlled temperature environment around the electrospinning setup is crucial to maintain the polymer in a molten state throughout the process. The molten polymer jet solidifies upon cooling, and the fibers are collected on a grounded substrate. The solidification process in melt-electrospinning is typically faster than in solution-based electrospinning due to the rapid cooling of the molten polymer, resulting in the formation of fibers with different morphological characteristics, such as increased crystallinity and mechanical strength.

Melt-electrospinning, while offering significant advantages, also has some limitations. The method eliminates the need for solvents, avoiding the associated health, safety, and environmental issues. It is an attractive method for producing nanofibers for applications requiring high purity and environmental safety. The fibers produced by melt-electrospinning often exhibit increased crystallinity and mechanical strength due to the rapid cooling and solidification of the molten polymer. However, the need to maintain the polymer in a molten state requires high temperatures, which can limit the range of polymers that can be processed and may degrade temperature-sensitive materials. The equipment needed for melt-electrospinning, including heated spinnerets and controlled-temperature environments, is more complex and expensive than solution-based electrospinning. The high viscosity of molten polymers can make achieving continuous and uniform fiber formation challenging, requiring precise control of operational parameters. Melt-electrospinning is particularly suitable for thermoplastic polymers, expanding the range of materials that can be processed into nanofibers.

To date, this process for producing PEDOT nanofibers has yet to be reported in the literature. The main reason is that PEDOT is usually available as a dispersion in water. Therefore, the most common way to fabricate nanofibers is using solution-electrospinning.

### 3.3. Coaxial Electrospinning

Coaxial electrospinning is a variation of the traditional electrospinning process, where two or three polymer solutions are simultaneously electrospun through a coaxial needle [33,34]. This method enables the fabrication of core–shell nanofibers with PEDOT as either the core or the shell material. Coaxial electrospinning is particularly useful for creating nanofibers with enhanced functionalities, such as improved electrical conductivity, mechanical properties, or biocompatibility. Preparing coaxial solutions involves selecting compatible core and shell materials with appropriate viscosities and conductivities. PEDOT:PSS can be combined with various polymers, such as poly(vinyl alcohol) (PVA) or polycaprolactone (PCL), to form core–shell structures. The core and shell solutions are loaded into separate syringes and fed through a coaxial needle, where they are electrospun together to form composite nanofibers. The electrospinning process for coaxial electrospinning is like traditional electrospinning but requires precise control of the flow rates for both the core and shell solutions. The applied voltage and working distance must also be optimized to ensure the formation of continuous and uniform core–shell fibers. The coaxial structure provides additional functionality to the nanofibers, such as enhanced conductivity from the PEDOT core or improved mechanical properties from the polymer shell. The core–shell nanofibers are collected on a grounded substrate, and the morphology and properties of the fibers can be tailored by adjusting the electrospinning parameters. Coaxial electrospinning offers a versatile approach to creating multifunctional nanofibers with potential applications in electronics, sensors, and biomedical devices.

Coaxial electrospinning allows the production of core–shell nanofibers, combining the properties of different polymers in a single fiber. This can enhance the functionality of the nanofibers, such as improving electrical conductivity, mechanical strength, or biocompatibility. By selecting different materials for the core and shell, the properties of the nanofibers can be finely tuned for specific applications. This method enables the creation of nanofibers with unique and enhanced characteristics. For example, coaxial electrospinning was employed by Olive et al. as a one-step alternative to fabricating nano-reinforced polyvinylidene fluoride (PVDF) fibers. This method enabled the dispersion of iron oxide (Fe_3_O_4_) nanoparticles with different size distributions (monomodal or bimodal) within the polymeric fibers (Figure 3). Coaxial electrospinning can encapsulate active ingredients, such as drugs or bioactive molecules, within the core of the fibers, protecting them from the environment and controlling their release. Like the previously discussed processes, coaxial electrospinning presents some limitations: the instrument setup is more complex than traditional electrospinning, requiring precise alignment and control of multiple polymer solutions. This increases the complexity and cost of the process. Achieving the desired core–shell structure requires careful optimization of the flow rates, viscosities, and conductivities of both solutions. Inconsistent parameters can lead to irregularities in the fiber structure. Finding solvents mutually compatible with the core and shell materials can be challenging, potentially limiting the range of materials used in coaxial electrospinning.

## 4. Main Characterization Techniques for PEDOT Nanofibers

Characterizing PEDOT nanofibers is essential for understanding their morphology, structure, and properties, directly influencing their performance in various applications. This section outlines the common techniques used for characterizing PEDOT nanofibers, including scanning electron microscopy (SEM), Transmission Electron Microscopy (TEM), X-ray Diffraction (XRD), Fourier Transform Infrared Spectroscopy (FTIR), and conductivity measurements. Nevertheless, all the characterization techniques used in materials science can be applied to investigating nonwoven materials. Each method provides unique insights into the physical and chemical attributes of the nanofibers, enabling comprehensive analysis and optimization for specific applications.

### 4.1. Scanning Electron Microscopy (SEM)

Scanning electron microscopy (SEM) is a powerful technique used to analyze the surface morphology and topography of PEDOT nanofibers. SEM operates by scanning a focused beam of electrons across the sample surface, which generates secondary electrons that are detected to produce high-resolution images. SEM provides detailed images of the nanofiber diameter, distribution, and surface texture. It allows for the observation of fiber alignment, uniformity, and the presence of any defects or bead formation. SEM images help us understand the effect of electrospinning parameters on fiber morphology, which is crucial for optimizing the process.

For instance, Zarrin et al. (2018) used SEM to analyze the morphology and diameter distribution of electrospun PEDOT nanofibers, while Costa et al. (2023) used SEM to obtain information about the thickness of nanofibers mats before and after vapor treatment with DMSO [31,35].

### 4.2. Transmission Electron Microscopy (TEM)

Transmission Electron Microscopy (TEM) involves transmitting a beam of electrons through a thin sample. The interaction of electrons with the sample produces an image that reveals internal structures at the nanoscale. TEM offers insights into the internal structure and crystallinity of PEDOT nanofibers. It can reveal the presence of core–shell structures, crystallites, and any phase separation within the fibers. TEM is particularly useful for understanding the distribution of conductive domains within the fibers, which impacts their electrical properties. Zarrin et al. (2018) used TEM to examine the protrusions of PEDOT nanofibers and related them to the surface roughness of nanofibers observed with SEM [31]. Examples of SEM and TEM images are reported in Figure 4.

### 4.3. X-ray Diffraction (XRD)

X-ray Diffraction (XRD) is a crucial tool for understanding the crystalline structure of PEDOT nanofibers. When X-rays are directed at a sample, they are diffracted by the crystal lattice, and the resulting diffraction pattern provides information about the lattice parameters and crystallinity. This analysis reveals the degree of crystallinity and the presence of any crystalline phases in the nanofibers. Such information is critical for understanding how the electrospinning process and subsequent treatments (e.g., annealing) affect the crystalline structure and, consequently, the electrical and mechanical properties of the nanofibers. Lim et al. (2023) employed XRD to study the crystallinity of PVDF-PEDOT:PSS nanofibers and found that the amount of PEDOT influences the relative quantity of the beta phase in PVDF [36].

### 4.4. Fourier Transform Infrared Spectroscopy (FTIR)

Fourier Transform Infrared Spectroscopy (FTIR) is a key technique for identifying chemical modifications in PEDOT nanofibers. It measures the absorption of infrared radiation by the sample as a function of wavelength, providing information about the chemical bonds and molecular structure of the material. FTIR analysis of PEDOT nanofibers identifies functional groups and confirms the chemical structure of the polymer. It can detect any chemical modifications, such as doping or the presence of additives, and monitor the interaction between PEDOT and other components in composite nanofibers. Huang et al. (2013) and later Verpoorten et al. (2020) used Differential Scanning Calorimetry and FTIR, respectively, to investigate the chemical modifications in the PEDOT:PSS-PEO system upon thermal treatment and associated it with the cross-linking between PSS and PEO chains [6,37].

### 4.5. Conductivity Measurements

Conductivity measurements assess the electrical properties of PEDOT nanofibers. Techniques such as four-point probes and impedance spectroscopy are commonly used to measure the electrical conductivity and impedance of the nanofiber mats or individual fibers. Conductivity measurements provide quantitative data on the electrical performance of PEDOT nanofibers. For example, Babaie et al. (2020) measured the electrical conductivity of cross-linked scaffolds [10]: three samples, each measuring 10 × 1 mm^2^, were cut from each electrospun sheet, and the thickness d of each sheet was recorded. The resistance (*R*) of nanofibers was determined by measuring the current while applying a voltage range from −10 V to 10 V in 0.2 V steps with a distance *l* between the electrodes (width *w* was 1 mm). The electrical conductivity *σ* of the fibers was calculated using Equation (1).
(1)$$\sigma =\frac{l}{wdR}$$

These measurements are essential for evaluating the effect of electrospinning parameters, post-treatment processes (e.g., solvent annealing), and the incorporation of conductive additives on the overall conductivity of the nanofibers. Park et al. (2016) conducted conductivity measurements on PEDOT:PSS nanofibers with PEO or PVA as the carrier polymer treated with DMSO and EG and observed a significant increase in conductivity when using PEO and EG thanks to the extended quinoid structure of the nanofibers [18]. Another common characterization of conductive nanofiber-based sensors is cyclic voltammetry, during which the electrochemical reaction of the system can be investigated as a function of the applied potential. For example, Barroso et al. (2023) reported that a foaming process has beneficial effects in terms of active surface area and porosity of sensors, improving their electrochemical response [13].

## 5. Properties of PEDOT Nanofibers

### 5.1. Electrical Properties

PEDOT nanofibers are known for their excellent electrical conductivity, which stems from the conjugated backbone of the PEDOT polymer. The conductivity of PEDOT nanofibers can range from 1 to 1000 S/cm, depending on factors such as the dopant used, the degree of crystallinity, and post-treatment processes like solvent annealing or thermal treatment [18,38]. Including dopants such as PSS (poly(styrene sulfonate)) or small molecules like ethylene glycol significantly enhances conductivity by increasing the carrier density and mobility [24]. Treatments such as solvent annealing with DMSO or ethylene glycol can remove excess insulating components (e.g., PSS) and improve the ordering of the PEDOT chains, leading to higher conductivity. For example, Costa et al. evaluated electrospun nanofibers of PEDOT:PSS and Ag nanowires chemically treated with DMSO vapor [35]. After the treatment, they observed an improvement of two orders of magnitude in the electrical conductivity of the film. The applied voltage, solution concentration, and flow rate during electrospinning affect the fiber diameter and uniformity, which in turn influence the overall conductivity of the nanofiber mat. High conductivity makes PEDOT nanofibers suitable for applications in flexible electronics, sensors, and energy storage devices like supercapacitors and batteries [5,11]. Their use in transparent conductive films and organic photovoltaics is also significant due to their excellent electrical properties and optical transparency [38]. Cyclic voltammetry (CV) and electrochemical impedance spectroscopy (EIS) are common electrical characterizations that can reveal the redox behavior of materials in nonwoven films by showing the current response as a function of the applied potential [39]. Peaks in the CV curves correspond to oxidation and reduction processes, providing insights into the electroactive species present in the film (Figure 5). The shape and symmetry of the CV curves indicate the reversibility of redox reactions. Sharp, symmetric peaks suggest reversible reactions, while broad or asymmetric peaks indicate quasi-reversible or irreversible processes. By cycling the potential multiple times, CV can assess the stability of nonwoven films under electrochemical conditions. Stable films will show consistent peak currents and shapes over successive cycles. The area under the CV curve can be used to estimate the capacitance of the film, which is crucial for applications in energy storage devices like supercapacitors. The current response at different scan rates can help distinguish between capacitive and faradaic (battery-like) behaviors. EIS measures the impedance of a system over a range of frequencies, providing detailed information about the charge transfer resistance at the interface between the nonwoven film and the electrolyte. This resistance is a critical parameter for evaluating the efficiency of electron transfer processes in devices like sensors and batteries. The impedance data can be used to determine the double-layer capacitance at the film–electrolyte interface. This capacitance is indicative of the surface area and the ability of the film to store charge. EIS can reveal information about ion diffusion and mass transport within the nonwoven film. Features such as the Warburg impedance, which appears at low frequencies, are associated with diffusion-controlled processes. The intrinsic electrical conductivity of the nonwoven film can be extracted by fitting the EIS data to equivalent circuit models. This includes separating contributions from bulk resistance, grain boundary resistance, and other interfacial resistances. When used together, CV and EIS can provide a comprehensive understanding of the electrochemical behavior of nonwoven films. CV can identify active redox processes, while EIS can quantify the resistive and capacitive components associated with these processes.

### 5.2. Mechanical Properties

PEDOT nanofibers typically exhibit moderate mechanical strength and flexibility. Their tensile strength ranges from 10 to 100 MPa, while the elastic modulus can vary widely based on the fabrication method and the presence of reinforcing agents [41]. Thinner fibers tend to have higher tensile strength due to the reduction in defects. Uniform fiber morphology also contributes to enhanced mechanical properties. Blending PEDOT with other polymers (e.g., PVDF, PVA, PLA) or incorporating nanofillers (e.g., carbon nanotubes, graphene) can significantly improve the mechanical strength and elasticity of the nanofibers. The solution viscosity, polymer concentration, and solvent type used during electrospinning affect the fiber formation process and, thus, the mechanical properties of the final nanofibers. For example, adding various concentrations of PEDOT:PSS to PVDF solutions has a direct impact on the deposition area, the diameter of the fiber, and the crystallinity of PVDF, as reported by Lim et al. The mechanical properties of PEDOT nanofibers are crucial for their use in wearable electronics, where flexibility and durability are essential [6,36]. They are also important in biomedical applications such as tissue engineering and drug delivery, where the mechanical strength of the scaffold influences cell attachment and growth [10].

### 5.3. Thermal Properties

PEDOT nanofibers exhibit good thermal stability, typically decomposing above 200 °C. The molecular structure and the presence of dopants or additives influence the thermal properties. Higher crystallinity and molecular weight generally enhance the thermal stability of PEDOT nanofibers. Certain dopants can either stabilize or destabilize the polymer matrix. For instance, PSS tends to increase the thermal stability of PEDOT due to strong ionic interactions. Incorporating thermally stable materials like inorganic nanoparticles can further improve the thermal stability of PEDOT nanofibers. The thermal stability of PEDOT nanofibers is critical for applications in high-temperature environments such as in sensors and actuators. It also plays a role in the long-term reliability of electronic devices, where thermal degradation can affect performance.

## 6. Applications of PEDOT Nanofibers

PEDOT (poly(3,4-ethylenedioxythiophene)) nanofibers have garnered significant attention across various fields due to their exceptional electrical, mechanical, and thermal properties. These properties make PEDOT nanofibers versatile materials suitable for energy storage, sensors, actuators, and tissue engineering. Recent research and developments in these application areas highlight the innovative use of PEDOT nanofibers and their potential to revolutionize technology.

### 6.1. Sensors

#### 6.1.1. Strain and Pressure Sensors

The applications of conventional sensors are limited by their boundedness, which is dependent on hard materials, uncontrolled self-multilayer structure building, and dense substrates, which are not ideal in terms of stretchability and permeability. The exceptional stretchability and air permeability of elastomeric nanofibers are attributed to their network structure. Stretchable conductors may be efficiently produced by elastomeric nanofibers by the introduction of metal nanofillers, intrinsic conductive polymers, carbon materials, and other techniques. This shows significant potential in the realm of flexible sensors [42]. Recently, the characteristics of composite solutions with poly(vinylidene fluoride) added to PEDOT:PSS were investigated, and the changes in electrospinning behavior and morphology were analyzed accordingly [36]. The research paper discusses the application of PVDF, a piezoelectric polymer, in various fields due to its flexibility and high piezoelectric constant. It also explores the potential application of PEDOT:PSS, a conductive polymer, in improving piezoelectric performance when combined with a piezoelectric polymer. The study suggests that the findings could lead to replacing ceramic-based materials with enhanced polymer-based piezoelectric elements, thereby accelerating the adoption of wearable piezoelectric devices. The methods used in this paper involved investigating the characteristics of composite solutions with PEDOT:PSS added to PVDF, analyzing changes in electrospinning behavior and morphology, and evaluating the crystallinity of PVDF in deposited fibers using Fourier Transform Infrared Spectroscopy and X-ray Diffraction analysis. Additionally, the study involved analyzing the changes in the radius of the jet, deposition area, fiber diameter, and crystallinity of the β phase as the PEDOT:PSS content increased, along with investigating the piezoelectric performance of the PVDF/PEDOT composite webs with various PEDOT:PSS contents [43]. Ding et al. showcase a method for producing mechanically stretchable and electrically conductive nonwoven textiles made of polyurethane (PU) through electrospinning and dip-coating with the conducting polymer PEDOT:PSS [44]. By adjusting the number of dip-coating times, nonwovens with initial sheet resistance ranging from 35 to 240 Ω/sq (electrical conductivity of 30–200 S m^−1^) can be conveniently prepared. The resistance of the nonwoven PEDOT:PSS@PU stabilizes after the first 10 stretch–release cycles, enabling its use as a stretchable conductor within a certain strain range for applications such as electric circuits.

Traditionally, integrating different sensors into one platform involves complex fabrication processes. Verpoorten et al. explored a simplified solution using a single piezoresistive and electrochemically active electrospun nanofiber mat [45]. This material serves as the sensitive element in a wearable physiological flex-sensing platform, reducing the process flow to just two steps. The resulting NFs pH-Flex Sensor can monitor both human joint deformation and skin pH. The pH sensing utilizes EIS, showing linear relationships between pH value and both double-layer capacitance and charge transfer resistance. Additionally, the sensor’s electrical resistance varies linearly with bending deformation. The gauge factors, 45.84 in traction and 208.55 in compression, indicate the nanostructured NFs’ remarkable piezoresistive behavior.

Recent studies have focused on thermoplastic polyurethane (TPU) as a primary material for flexible strain sensors [46,47,48]. TPU is highlighted for its mechanical flexibility and compatibility with various conductive materials. Different conductive fillers, such as carbon black (CB), carbon nanotubes (CNTs), graphene flakes (GRs), MXene, metallic materials, and conductive polymers like PEDOT have been incorporated inside TPU. These fillers enhance the electrical conductivity and mechanical properties of the composite materials. The use of electrospinning is emphasized for fabricating nanofiber composites. This method enables the creation of highly stretchable and conductive fibers, essential for developing strain sensors with high sensitivity, stretchability, and a wide sensing range. The effectiveness of these sensors is discussed in various applications, including motion detection and biomedical uses. Each study meticulously analyzes the conductivity, mechanical properties, and strain-sensing performance of the composites (see Figure 6), exploring the effects of filler dimensionality and synergy, as well as the structural evolution and re-arrangement of filler networks under mechanical stress.

#### 6.1.2. Chemical Sensors

The high surface area and conductivity of PEDOT nanofibers make them ideal for chemical sensing applications, where they can detect gases, ions, and biomolecules [13]. Huang et al. (2015) achieved the controlled and continuous electrospinning of nearly pure PEDOT nanofibers by using magnesium nitrate as a cross-linker to enhance polymer entanglement [49]. Compared to bulky PEDOT films, these electrospun nanofibers demonstrated significantly higher sensitivities and faster response times in detecting gas molecules of dimethylformamide, dimethyl sulfoxide, and propylene carbonate. Shiu et al. (2022) recently developed a gas sensor for ethanol monitoring [50]. In their experiment, a gas sensitivity measurement system (WS-30B) was utilized to evaluate the gas sensitivity of PVP/PEDOT/TiO_2_ nanofiber membranes. The sensor operated with a voltage of 5 V, and a steady baseline was established between 0 and 1. After two minutes, the specified gas or liquid was introduced, and the detector measured the change in electrical resistivity. Ethanol was efficiently evaporated using a heater, and the resulting gas was evenly diffused throughout the test case with a blowing air set. Tests were conducted at room temperature, with air as the carrier gas, to simulate a typical sensor environment. The materials’ gas sensitivity and its relationship to relative humidity were assessed by comparing the electrical resistivity of the sensor before and after exposure to the test conditions (Figure 7). They observed a higher sensitivity for electrospun nanofiber-based sensors than other fabrication techniques (like PEDOT in situ polymerization). Regmi et al. investigated the hydrogen gas sensing mechanism of suspended graphene/PEDOT:PSS/PEO composite nanoscale channels created through near-field electrospinning [51]. Due to effective charge transfer at the graphene–PEDOT:PSS-PEO interface, these channels offer a higher surface area-to-volume ratio for efficient gas diffusion and enhanced response. The sensor achieves a 2% response to 1 ppm H_2_ concentration with good linearity across a wide dynamic range. Additionally, arrays of nanoscale channels for improved sensitivity and integrated microheaters for rapid device recovery are implemented.

Electrospun PEDOT:PSS/PVP composite nanofibers have been successfully used for sensing CO [52].

Numerous studies have explored the use of conducting polymers like polypyrrole, polythiophene, and polyaniline for gas sensor applications. In recent years, there has been a growing interest in poly(3,4-ethylenedioxythiophene)-poly(styrenesulfonate) (PEDOT:PSS) as a sensing material due to its exceptional electrical conductivity and environmental stability. Additionally, PEDOT:PSS/insulating polymer composites exhibit responsiveness to chemical vapors. For instance, electrospun polyvinyl alcohol/PEDOT:PSS (PVA/PEDOT:PSS) fibers have been developed for the detection of formaldehyde, ethanol, and other organic gases using the resistance measuring method. These fibers possess high porosity and specific surface area, offering improved sensitivity and faster response time compared to thin-film sensors.

Zhang et al. conducted a study involving creating conductive fibers by adding PEDOT:PSS conductive water solution into a 10% PVA water solution [53]. Electrospinning experiments were conducted under high-pressure airflow assistance, with the maximum spinning voltage set at 70 kV and a receiving distance of 120 cm. As a comparison, PVA/PEDOT:PSS composite fibers were also prepared using a traditional electrospinning device with a spinning voltage of 20 kV and a receiving distance of 15 cm. The primary objective was to investigate the impact of the specific surface area (size effect) of electrospun fibers on their response to external stimuli. Comparative tests were carried out on ultrafine nanofiber membranes and traditional electrospun fiber membranes to evaluate the response characteristics of PVA/PEDOT:PSS fibers, focusing on changes in electrical resistance in an ammonia environment. The study observed and discussed the size effect of electrospun fibers on gas-sensing properties such as sensitivity and response time.

J. Choi et al. fabricated PEDOT:PSS/PVP nanofibers via electrospinning, and the verification of the presence of PEDOT:PSS in the nanofibers was achieved through FT–Raman spectroscopy analysis [54]. Furthermore, the detection capabilities of the electrospun PEDOT:PSS/PVP nanofibers were investigated by assessing their reaction to repeated exposure to organic vapors like ethanol, methanol, THF, and acetone at ambient temperature. The exposure of PEDOT:PSS/PVP nanofibers to different solvents demonstrated contrasting electrical behaviors between protic and aprotic solvents. These outcomes suggest that the electrospun PEDOT:PSS/PVP nanofibers hold substantial promise as a material for sensing organic vapors.

#### 6.1.3. Biosensors

PEDOT nanofibers are employed in biosensors to detect biological molecules such as glucose, DNA, and proteins, owing to their biocompatibility and conductive nature. For example, combining PEDOT with carbon nanofibers (CNFs) resulted in a novel composite material exhibiting low impedance, high surface area, high charge injection capability, and efficient neurotransmitter monitoring, making it highly suitable for neuronal therapies [55]. The PEDOT-modified microelectrodes showed superior electrochemical performance, with a low specific impedance of 1.28 MΩ μm^2^ at 1 kHz and a high charge injection limit of 10.03 mC/cm^2^, while demonstrating excellent sensitivity and low detection limits for dopamine and serotonin, and no cytotoxicity, highlighting PEDOT’s potential for next-generation neural microelectrodes. Meng et al. (2020) present a method for creating a bi-functional PEDOT interface with a 3D nanofibrous network and carboxylic acid groups, enabling the development of all-polymer-based biosensors [56]. The Nano-PEDOT-COOH, with tunable fiber diameters and carboxylic acid group densities, showed excellent electrochemical properties and the capability for stable enzyme immobilization, resulting in a highly sensitive and responsive biosensor for lactate detection.

Electrospun PEDOT:PSS-based nanofibers have been developed for the fabrication of wearable biosensors for dopamine and ascorbic acid [45,57]. The nanofibers showed excellent biocompatibility and electrical conductivity, making them suitable for biosensing applications.

### 6.2. Biomedical

PEDOT exhibits several advantageous properties that make it highly beneficial for biomedical applications. Its excellent electrical conductivity and biocompatibility make PEDOT an ideal material for neural interfaces [55], where stable and reliable electrical performance is critical for stimulating and recording neuronal activity. The biocompatibility of PEDOT helps minimize inflammatory responses and tissue damage, which is crucial for long-term implantation [58]. Additionally, PEDOT’s flexibility allows it to conform to the complex geometries of biological tissues, enhancing the interface between the device and the tissue [59]. This flexibility also makes PEDOT suitable for wearable biomedical devices, where it can maintain its conductive properties under mechanical deformation (Figure 8). Furthermore, PEDOT can be fabricated into various forms, such as thin films and electrospun nanofibers, enabling its use in several applications, from biosensors to biofuel cells. These properties collectively make PEDOT versatile and valuable in advancing biomedical technologies.

For example, electrospun PEDOT:PSS-based nanofibers showed good biocompatibility and electrical conductivity, making them suitable for biomedical devices like biosensors and neural implants [61]. In another study, electrical conductive composite nanofibers were fabricated using PEDOT:PSS with cellulose nanofibrils (CNFs) [62]. The fabrication process involved electrospinning the composite nanofibers to create a water-resistant and cytocompatible material. Recently, highly electrically conductive nanofibers made of poly [2,2′-m-(phenylene)-5,5′-bibenzimidazole] (PBI) have been manufactured by electrospinning and then coated with cross-linked poly (3,4-ethylenedioxythiophene) doped with poly (styrene sulfonic acid) by spin-coating or dip-coating [63]. The methods used in the paper include the manufacture of highly electrically conductive nanofibers made of poly [2,2′-m-(phenylene)-5,5′-bibenzimidazole] (PBI) by electrospinning. These nanofibers were then coated with cross-linked poly (3,4-ethylenedioxythiophene) doped with poly (styrene sulfonic acid) (PEDOT:PSS) using spin-coating or dip-coating techniques. Characterization of the scaffolds was carried out through scanning electron microscopy (SEM) imaging and attenuated total reflectance Fourier Transform Infrared (ATR-FTIR) Spectroscopy. The electrical conductivity of the fibers was measured using the four-probe method, showing significant increases in conductivity for both spin-coated and dip-coated samples. Experimental parameters were carefully explored to achieve reproducible conductive nanofibers synthesis in large quantities, focusing on the critical importance of relative humidity during the electrospinning process and doping post-treatment involving glycols and alcohols. The synthesized fibers were assembled as a mat on glass substrates to form a conductive and transparent electrode, which was then fully characterized for its optoelectronic properties.

#### 6.2.1. Tissue Engineering

The biocompatibility and conductive properties of PEDOT nanofibers make them suitable for scaffolds in tissue engineering, where they can support cell growth and differentiation [10]. For example, Abedi et al. (2019) investigated chitosan scaffolds with varying concentrations of PEDOT for cardiac tissue engineering [64]. The findings indicate that the addition of PEDOT enhances the scaffolds’ mechanical properties, electrical conductivity, biocompatibility, and cell viability, with the 1 wt% PEDOT scaffold showing a significant 30–40% reduction in fiber diameter, a 100-fold increase in electrical conductivity, and a tensile strength increase of about 9 MPa, making it a promising candidate for cardiac tissue engineering applications. In another study, Jin et al. (2013) developed PEDOT nanofiber mats with superior mechanical properties (tensile strength: 8.7 ± 0.4 Mpa; Young’s modulus: 28.4 ± 3.3 Mpa) and flexibility, alongside high biocompatibility comparable to tissue culture plates when tested with human cancer stem cells [17]. With outstanding electrical conductivity (7.8 ± 0.4 S cm^−1^), these PEDOT nanofiber mats are promising for various biotechnology applications, including tissue engineering, drug delivery, cell culture, and implanted electrodes.

Electroconductive biomaterials have emerged to facilitate the recovery of degenerated electrically conductive tissues, particularly cardiac tissues following myocardial infarction. F. Furlani et al. investigated the fabrication of electroconductive scaffolds for the regeneration of cardiac tissue utilizing the biocompatible and conductive polymer PEDOT:PSS, in conjunction with a biomimetic gelatin polymer network [65]. The utilization of dehydrothermal (DHT) treatment under vacuum conditions produced structurally stable scaffolds without the need for additional cross-linking agents. These resultant scaffolds emulate the Young’s modulus of native cardiac tissue and possess a well-interconnected porosity, along with excellent swelling capacity and stability under physiological conditions. Moreover, the incorporation of PEDOT:PSS serves to augment the electroconductivity of the resulting materials. All scaffolds exhibit non-cytotoxicity towards H9c2 cardiomyoblasts, with PEDOT:PSS promoting cell adhesion, particularly in the initial stages, essential for favorable outcomes post-implantation, as well as cell proliferation and spreading on the scaffolds. Given the favorable interaction of the scaffolds with cardiomyoblasts, these biomimetic and electroconductive scaffolds demonstrate potential as implantable biomaterials for regenerating electroconductive tissues, especially cardiac tissue, and as a promising 3D tissue model for in vitro screening of biomolecules.

#### 6.2.2. Drug Delivery Systems

PEDOT nanofibers can be employed in drug delivery systems due to their ability to load and release therapeutic agents in a controlled manner. Chen et al. (2017) introduce multifunctional core–shell hybrid microfibers for biomedical applications, synthesized using coaxial spinning and dip-coating [66]. These microfibers are composed of a core made of bacterial cellulose and a shell layer of PEDOT, offering improved mechanical characteristics, controlled drug release, and responsiveness to electrical stimulation, demonstrating potential for applications in tissue engineering and drug delivery. 

### 6.3. Wearable Electronics

The absence of mechanically strong fibers and yarns with good electrical conductivity impedes the development of electronic textiles. Conjugated polymer fibers show great promise in fulfilling those demands [67]. Electrospun composite nanofibers of PEDOT:PSS/cellulose acetate and PEDOT:PSS/PEO have been studied for electromagnetic interference (EMI) shielding [68]. The nanofibers showed excellent EMI shielding effectiveness and mechanical properties suitable for electronic devices. The EMI shielding is based on reflection and absorption loss. The former is the weakening of the electromagnetic wave when passing through materials with mismatching dielectric constants. The latter is the weakening when propagating inside a material. The reflection loss is enhanced in nanofiber-based materials due to the large surface to volume ratio and can be further improved with coaxial or multiwire fiber geometries. Absorption loss can be achieved if the thickness of the fabric is sufficient and can be improved by adding nanofillers with magnetic properties to the fibers.

Stretchable conductors and organic electrochemical transistors (OECTs) were fabricated from PEDOT:Tos (poly (3,4-ethylenedioxythiophene):iron tosylate) nanofibers [69]. The devices were prepared by a combination of electrospinning and electrode printing followed by vapor-phase polymerization (VPP). The impact of both the processing time and the composition of three electrospinning mixtures on the electrospun fiber mats was evaluated by scanning electron microscopy and cyclic voltammetry. Fibrillar mats prepared from the different mixtures maintained their electrical properties and could be stretched up to 140% of their original length. Stretchable OECTs were fabricated by printing silver drain and source electrodes directly on the conductive spun fibers. The fabricated devices showed transistor behavior up to ∼50% strain.

Novel evidence is shown on the capability of tuning the conductance and photoresponse of composite core–shell nanofibers, based on the doping of the PEDOT:PSS phase with different solvents and PbS nanoparticles and the arrangement of the core–shell phases [70]. The study employed coaxial electrospinning to synthesize core–shell PEDOT:PSS–polyvinylpirrolidone nanofibers. The fibers were doped with different solvents such as dimethyl sulfoxide (DMSO), isopropyl alcohol (IPA), ethylene glycol, and PbS nanoparticles at varying concentrations. 

#### Flexible Soft Electronics

Recent advancements in flexible and conductive nanofiber materials highlight their potential in various applications, including wearable electronics, and in flexible and conductive materials [71]. With the expansion of the Internet of Things (IoT), a demand arises for portable and wearable devices that exceed the capacities of conventional silicon-based electronics, thereby leading to the emergence of novel nano-systems integrating conductive electrospun fibers. Furthermore, flexible and transparent materials represent alternatives to expensive and environmentally unsustainable electrodes based on indium and tin used in various fields, such as smart displays, energy, and wearables. However, few alternatives have all the desired properties. Due to the limitations of indium tin oxide, alternative electrodes like metal films and nanowires have emerged. Nanofibers are seen as an ideal material for high-performance transparent electrodes [72]. Electrodes of silver nanoparticles dispersed in electrospun PEDOT nanofibers have been studied for their high conductivity and mechanical flexibility, with a transparency of 77% [73]. A highly conductive nanocomposite created by grafting polyvinyl alcohol onto surface-modified carbon nanofibers demonstrated significant improvements in flexibility, conductivity, and biodegradability, making it suitable for biofuel cell electrodes and wearable electronics [60].

A recent work by Gramont et al. develops highly stretchable poly-(3,4-ethylenedioxythiphene) doped with tosylate (PEDOT:Tos) nanofibers that can be stretched up to 140% of the initial length, maintaining high conductivity [74]. The methods used in the paper involved a two-step process which included electrospinning of a carrier polymer with an oxidant and vapor-phase polymerization to produce highly stretchable poly-(3,4-ethylenedioxythiphene) (PEDOT) doped with tosylate (PEDOT:Tos) nanofibers on a polydimethylsiloxane substrate.

A process to synthesize continuous conducting nanofibers was developed using PEDOT:PSS as a conducting polymer and an electrospinning method, producing a conformable conductive and transparent coating that is well adapted to nonplanar surfaces, with very large aspect ratio features [75].

The rise of lightweight, flexible, and portable electronics and wearable devices has led to concerns over increased electromagnetic interference (EMI) levels. To mitigate this issue, the urgent development of lightweight, hydrophobic, and flexible EMI shielding materials is necessary.

Lee et al. developed core–shell nanofibers of PVDF with a highly conductive PEDOT shell for EMI shielding, showcasing the core–shell formation through vapor-phase polymerization [76]. The resulting nanofibers, known for flexibility and mechanical strength, exhibited an enhanced EMI SE of 40 dB in the X-band range due to increased internal reflections from porosity, with specific shielding effectiveness reaching 16,230 dB cm^2^ g^−1^ at a 14 μm thickness, attributed to the PEDOT shell’s high conductivity, durability, and hydrophobic properties.

Through the fabrication of poly(3,4-ethylenedioxythiophene): polystyrene sulfonate (PEDOT:PSS) and polyvinylpyrrolidone (PVP)-coated, carbonized electrospun polyacrylonitrile (PAN) nanofibers (CNFs) and their flexible polydimethylsiloxane (PDMS) composites, a highly effective EMI shielding material with a shielding effectiveness of 44 dB in the frequency range of 8–26.5 GHz and an absolute EMI SE of 5678 dB cm^2^ g^−1^ has been created [77]. The shielding mechanism of PEDOT:PSS-PVP/CNF composites is primarily absorption-based, highlighting their potential as practical applications for thin, flexible, hydrophobic, and lightweight EMI shielding materials.

### 6.4. Energy Storage

PEDOT nanofibers are widely used in supercapacitors due to their high electrical conductivity and large surface area [24]. Their fibrous morphology provides a porous structure, enhancing ion diffusion and storage capacity [78], as represented in Figure 9.

This enhancement can be explained in terms of the hierarchical substructure of the yarns and mats, which can be easily tuned to allow optimized electronic transportation and ion diffusion from the interior of the material to the surface by shortening the ion diffusion distance. The large surface area-to-volume ratio and the easy tunability of the fiber structure are indeed intrinsic advantages of the electrospinning technique. On the other hand, efficient electron charge transfer is enabled by the high conductivity of the PEDOT polymer.

PEDOT nanofibers produced through electrospinning and vapor-phase polymerization had a 350 nm diameter and high conductivity (60 ± 10 S cm^−1^), the highest reported for polymer nanofibers [79]. The mats, with soldered intersections, exhibited strong dimensional stability. These ultra-porous mats enhanced electrochemical properties. All-textile flexible supercapacitors integrated the mats with carbon cloths and PAN nanofibrous membranes. The supercapacitors were flexible, utilizing a solid electrolyte with an ionic liquid and PVDF-co-HFP. These devices displayed stable performance in ambient conditions.

A recent study by Du et al. (2022) presents a low-cost method for creating mechanically strong and conductive PEDOT bulk films using cellulose nanofibrils (CNFs) as building blocks [80]. The resulting PEDOT/CNF nano-paper exhibits excellent flexibility, high tensile strength (72 MPa), and high electrical conductivity (66.67 S/cm), making it a promising electrode material for flexible supercapacitors with outstanding cycling stability and high performance.

Finally, PEDOT:PSS electrodes were used in supercapacitors with polyvinyl alcohol/phosphoric acid (PVA/H_3_PO_4_) gel polyelectrolyte, forming an all-solid-state SC [17]. The methods employed in this paper involved fabricating flexible and transparent supercapacitors using electrospun PEDOT:PSS electrodes. These electrodes were prepared through electrospinning techniques to enhance their flexibility and transparency.

PEDOT nanofibers serve also as conductive additives in lithium-ion and sodium-ion batteries, improving charge transport and overall battery performance [81]. For example, in a recent work by Zeng et al. (2021), a network of carbon nanofibers (NCNF) was prepared to form a composite with sulfur for lithium–sulfur batteries but initially showed low specific capacity and poor cycle stability due to poor conductivity and large pore size [82]. Coating the NCNF with PEDOT significantly improved conductivity and addressed lithium polysulfide dissolution/loss, resulting in a composite electrode with a high sulfur loading that achieved a specific capacity of 793 mAh/g at 0.1 C, maintaining 623 mAh/g after 200 cycles, thus enhancing low current rate cyclability.

A. Laforgue created Poly(3,4-ethylenedioxythiophene)/manganese oxide coated on porous carbon nanofibers (P-CNFs/PEDOT/MnO_2_) through a combination of electrospinning, carbonization, and electrodeposition, forming an advanced anode material [83]. The structural and morphological analysis shows that the rough surface of the electrode provides active sites for Li+ storage due to the presence of PEDOT nanoparticles and irregular block-shaped MnO_2_. The electrode demonstrates superior electrochemical performance with a discharge capacity of 1477 mAh/g at 2 mA/g, outperforming other materials like P-CNFs/PEDOT (1191 mAh/g) and P-CNFs/MnO_2_ (763 mAh/g). After 20 cycles, the P-CNFs/PEDOT/MnO_2_ electrode maintains satisfactory performance with a Coulombic efficiency above 90% and a lower charge transfer resistance compared to other electrodes. Overall, the results suggest that this electrode has the potential to replace commercial graphite in lithium-ion batteries.

## 7. Challenges and Future Perspectives

Electrospun PEDOT-based nanofibers hold significant promise for various applications, yet they face several challenges related to scalability, reproducibility, stability, and integration into practical devices. 

Scalability is one of the primary challenges of the electrospinning process, which is often limited by the need for precise control over fiber morphology and uniformity. This issue can be addressed by optimizing the electrospinning parameters and developing advanced fabrication techniques that ensure consistent fiber production at a larger scale. Recent strategies include the use of multi-needle systems, auxiliary electrodes, and controlled environmental conditions, each contributing to overcoming specific limitations of traditional electrospinning, focusing on enhancing production capacity, and ensuring consistent fiber quality. The use of multiple electrospinning needles operating in parallel is a prominent strategy to increase production capacity. This approach, however, introduces challenges such as uneven electric field distribution due to Coulombic repulsion between charged jets and needles. To mitigate this, auxiliary electrodes have been employed to uniformize the electric field, thereby improving the deposition area and fiber morphology. Implementing parallel lateral plates as auxiliary electrodes has been shown to effectively address the uneven electric field distribution in multi-needle systems. This setup helps achieve a uniform electric field, which is crucial for consistent fiber production. The use of finite element simulations has further validated the impact of these electrodes on the electric field intensity, leading to high-quality polymer nanofibers [84].

Reproducibility is another critical concern, as variations in fiber properties can lead to inconsistent device performance. Enhancing reproducibility can be achieved by implementing standardized protocols and integrating real-time monitoring systems in the electrospinning process to guarantee consistency in the characteristics of the fibers. Reproducibility in electrospinning is significantly influenced by environmental conditions, particularly relative humidity (RH). A miniaturized electrospinning setup, designed to fit within a biological safety cabinet, allows precise control of RH, enhancing the reproducibility of fiber production. This setup demonstrated that controlling RH can significantly affect fiber diameter, thus ensuring consistent quality across production batches [85]. While these optimization strategies have advanced the scalability and reproducibility of electrospinning, challenges remain, particularly in balancing high throughput with fiber quality. Future research may focus on integrating these advanced electrospinning configurations with automated systems to further enhance production efficiency and consistency.

Real-time monitoring of nanofiber quality is conducted by means of interferometry: in industrial roll-to-roll setups, fiber thickness can be estimated with an optical interferometer, which provides feedback to the flow rate controller of the electrospinning instrument. Furthermore, a novel optical system integrated through a coaxial needle has recently been proposed to monitor the process, aiming to enhance the reproducibility and quality of nanofibers [86]. The system uses an optical fiber to pass light through the polymeric solution and couple it to the electrospun fiber. Experimental tests confirmed that the system effectively tracks changes in the electrospun fiber’s waveguide properties and nanofiber diameter, offering a promising solution for industrial-scale electrospinning using multi- or moving needles.

Stability, particularly in harsh environmental conditions, is essential for the practical application of these nanofibers. To the best of our knowledge, no specific study has been published on the stability of PEDOT-based nanofibers yet. Nevertheless, a few recently published studies have addressed the stability of PEDOT-based thin films investigating the aging/degradation mechanism and possible enhancement methods [87,88,89]. Such improvements could be obtained by surface modifications [90], stabilizing agents [88], or by coatings that protect the fibers from degradation due to moisture, temperature, and UV exposure [87,89].

Integrating practical devices, such as wearable sensors and electronic skins, requires the nanofibers to exhibit excellent mechanical properties, biocompatibility, and electrical performance. This can be facilitated by developing composite materials that combine PEDOT with other polymers or nanomaterials to enhance their mechanical strength and flexibility while maintaining their electrical properties. For conducting polymers like PEDOT:PSS, solution formulation improvements have been crucial. By optimizing the polymer concentration and solution properties, researchers have achieved significant enhancements in fiber conductivity and power factor, which are essential for applications in wearable electronics [26]. For example, stretchable conducting polymeric films were fabricated by blending PEDOT:PSS with poly(vinyl alcohol) in polar organic solvents like DMSO and EG [91], resulting in enhanced electrical and mechanical properties. The study found that DMSO and EG induce crystalline domains and structural changes in PEDOT, significantly boosting electrical performance and stretchability. It is reasonable to expect a similar behavior in nonwoven materials, too.

Future research should focus on developing multifunctional nanofibers that simultaneously address multiple performance criteria, such as conductivity, flexibility, and durability. Additionally, exploring advanced characterization techniques to understand the structure–property relationships in PEDOT-based nanofibers will be crucial for optimizing their performance in various applications. Adopting green and sustainable electrospinning processes will also be vital to minimize the environmental impact of large-scale production. In fact, this fabrication technique, which is a form of additive manufacturing, offers several environmental benefits. Firstly, electrospinning is highly efficient, allowing for precise control over material usage, thereby minimizing waste. The process typically operates at ambient temperatures, especially when carried out in a solution-based configuration, reducing energy consumption compared to traditional manufacturing methods. Additionally, the ability to utilize biodegradable or recyclable polymers in electrospinning further enhances its environmental credentials. Furthermore, traditional fiber production methods like Chemical Vapor Deposition (CVD), melt blowing, or solution blow spinning often require large amounts of solvents or other chemicals, which can lead to hazardous waste and environmental pollution. Electrospinning can use greener solvents or even solvent-free approaches, thereby minimizing chemical waste and the environmental impact. However, while the potential for sustainability is clear, comprehensive Life Cycle Assessment (LCA) studies are still needed to fully quantify the environmental impact of large-scale electrospinning and to identify areas for further improvement. By addressing these challenges through targeted research and development, PEDOT-based nanofibers can be effectively scaled up and integrated into a wide range of practical devices, facilitating widespread adoption in different fields, such as healthcare, environmental monitoring, and flexible electronics. Current wearable sensors often lack comfort and face issues with signal interference from different stimuli. Functional sensors that address these challenges are essential for providing reliable interaction between humans and their environment. PEDOT-based nonwovens are solid candidates for the development, for example, of flexible, stretchable electronic skins that mimic natural skin for integration in smart clothing [92,93].

## 8. Conclusions

The production of nanofibers has become a crucial area of study because of their distinct characteristics and diverse applications in various fields like biomedicine, textiles, energy, and environmental science. Electrospinning has gained notable attention as a method for creating nanofibers due to its adaptability, scalability, and capability to produce nanofibers with specific properties. It involves applying a high-voltage electrical field to a polymer solution or melt, forming a thin jet that elongates and solidifies into continuous nanofibers as the solvent evaporates or the polymer solidifies. These nanofibers have diameters ranging from a few nanometers to several micrometers, exhibiting exceptional uniformity and purity. The high surface area-to-volume ratio of electrospun nanofibers enhances their interactions with the surroundings, making them ideal for filtration, tissue engineering, drug delivery, and sensing applications. Additionally, the properties of these nanofibers can be easily adjusted by modifying parameters such as polymer concentration, solvent type, applied voltage, and collection distance, offering customization to meet specific needs across various fields. Besides their customizable properties, electrospun nanofibers are also scalable and cost-effective, suitable for industrial-scale production. The simplicity of the electrospinning process, along with its ability to produce uniform and pure nanofibers, presents an appealing option for large-scale manufacturing. Conductive polymers have gained attention for their unique combination of electrical conductivity, mechanical flexibility, and processability, opening new possibilities for advanced electronic devices and energy storage systems. Various experiments have been carried out on polymers like polyaniline (PANI), polypyrrole (PPy), polythiophenes, etc. Among these, poly(3,4-ethylenedioxythiophene) (PEDOT) is notable for its exceptional conductivity, environmental stability, and ease of synthesis. Belonging to the polythiophene family, PEDOT is characterized by its π-conjugated structure with alternating double and single bonds along the polymer backbone, providing high electrical conductivity. The presence of ethylenedioxy (EDOT) functional groups enhances the solubility and processability of PEDOT, facilitating the production of thin films, coatings, and composites. Electrospun PEDOT-based nanofibers have adjustable electrical and optical properties, suitable for applications in organic electronics, energy storage, biomedicine, and wearable technology. While electrospinning offers significant advantages for PEDOT-based conductive nanofibers, it also presents challenges such as the need for precise control over processing parameters to achieve desired fiber characteristics. Additionally, the inherent low molecular weight and rigidity of conductive polymers like PEDOT can complicate the electrospinning process, often requiring the use of carrier polymers or coaxial spinning techniques to achieve stable fiber formation. Despite these challenges, the versatility and adaptability of electrospinning make it a promising method for advancing the development of high-performance conductive nanofibers across various applications.

## Figures and Tables

**Figure 1 polymers-16-02514-f001:**
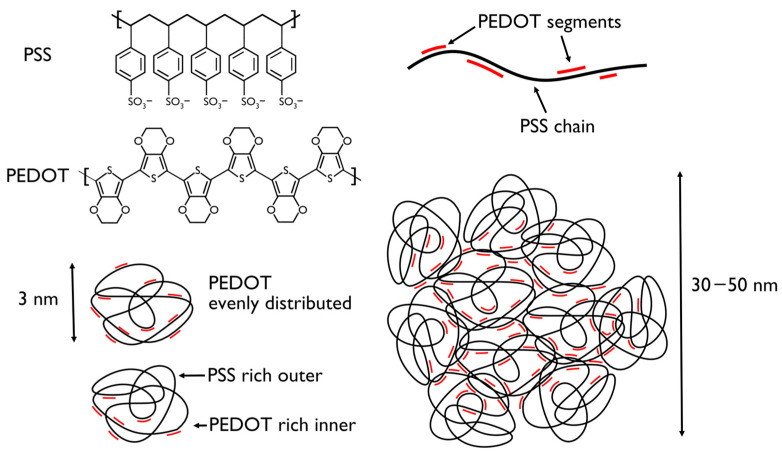
It is proposed that PEDOT:PSS grain consists of numerous tangles, where a single PSS chain forms each tangle with several PEDOT segments attached. These tangles are organized inside the grain so that the PEDOT segments are evenly distributed. On the outer layer of the grain, the tangles are arranged such that the hydrophilic PSS parts are oriented outward, allowing the grains to be dispersed in an aqueous solution. This arrangement facilitates the even distribution and dispersion of PEDOT grains in water.

**Figure 2 polymers-16-02514-f002:**
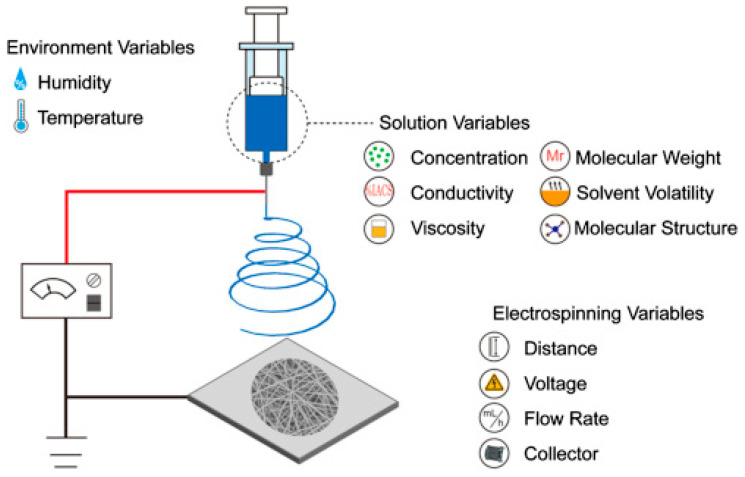
Schematic representation of a conventional electrospinning setup, alongside the environmental conditions, the characteristics of the solution utilized, and the various electrospinning parameters that can significantly influence the resultant fiber morphology and structure. An electrospinning system is made up of four main parts: a syringe pump, a voltage power supply, a needle, and a collector. Reproduced with permission from Long et al. in Electrospinning: Nanofabrication and Applications, published by Elsevier: Amsterdam, 2019 [23].

**Figure 3 polymers-16-02514-f003:**
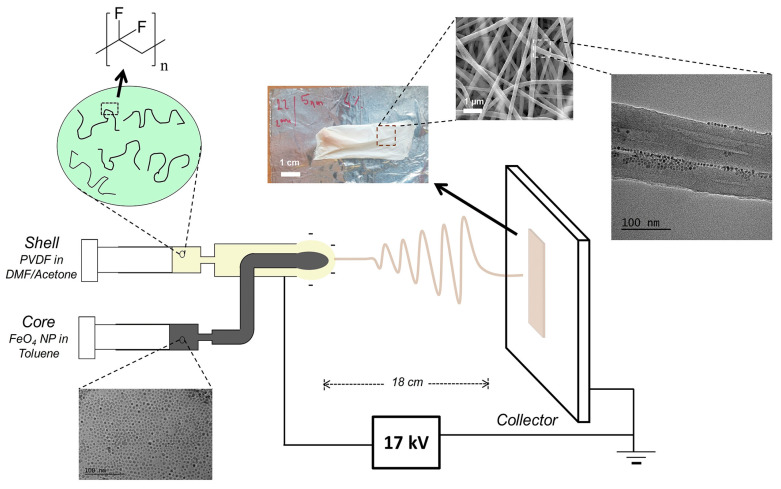
Schematic representation of the coaxial electrospinning process. Reproduced with permission from Oliva et al. in Polymer Composites, published by Wiley, 2021 [33].

**Figure 4 polymers-16-02514-f004:**
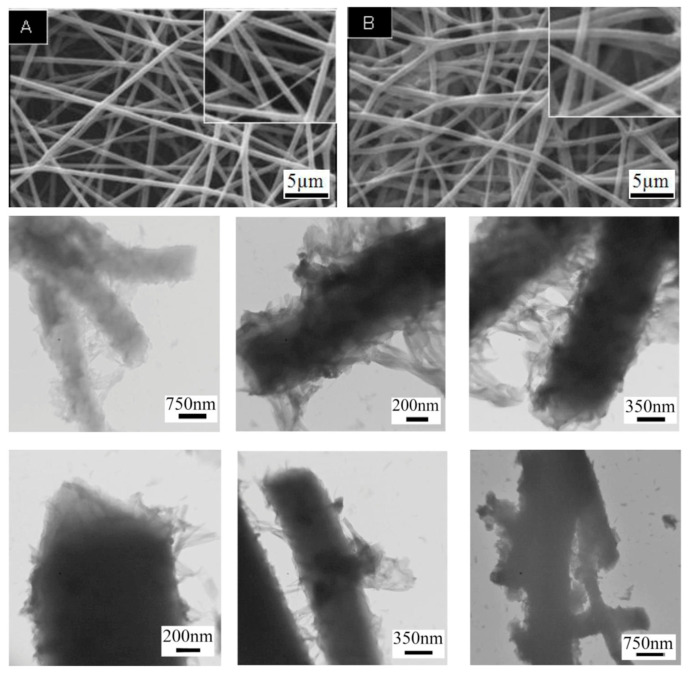
(Top) SEM micrograph and corresponding nanofiber diameter distribution are shown for (**A**) electrospun EDOT-PAA (67:33) and (**B**) electrospun PEDOT-PAA (67:33) directly oxidized in an oxidation bath containing FeCl_3_·6H_2_O. (mid and bottom). Typical TEM images of PEDOT-PAA (50:50) nanofibers are also presented. Reproduced with permission from Zarrin et al. in Synthetic Mets, published by Elsevier, 2018 [31].

**Figure 5 polymers-16-02514-f005:**
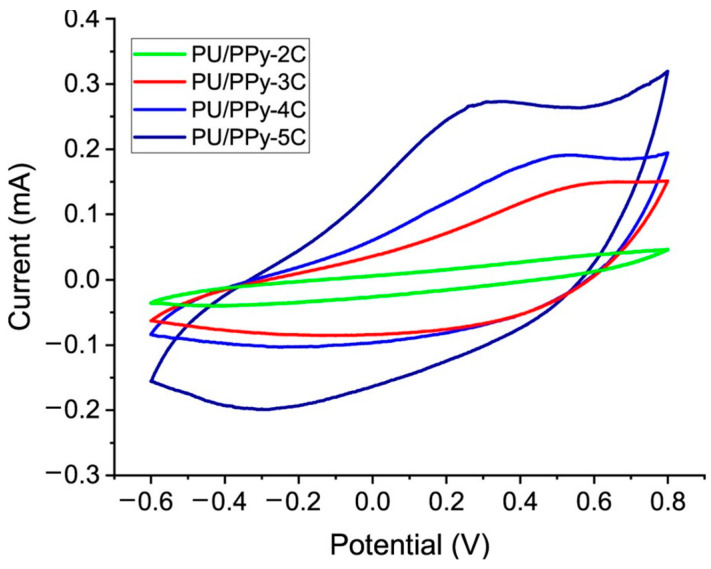
Cyclic voltammetry of PU/PPy nanofibrous actuators in the 0.1 M LiClO_4_ electrolyte solution between potentials of 0.6 to 0.8 V with a scan rate of 5 mV s^−1^. Reproduced with permission from Ebadi et al. in Smart Materials and Structures, published by IOP Publishing, 2020 [40].

**Figure 6 polymers-16-02514-f006:**
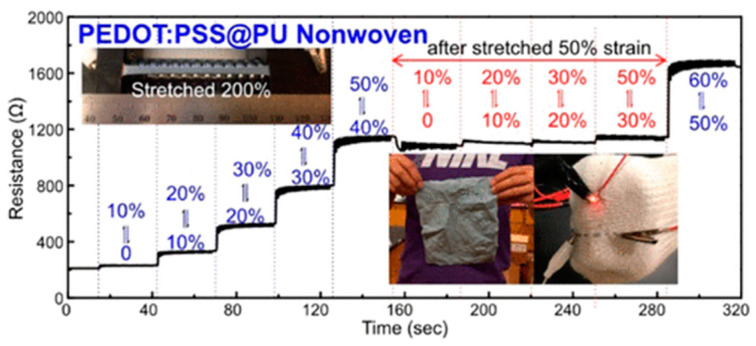
Resistance of the PEDOT:PSS/PU composite nonwoven as a function of stretching time. Reproduced with permission from Ding et al. in ACS Appl. Mater. Interfaces, published by American Chemical Society, 2017 [44].

**Figure 7 polymers-16-02514-f007:**
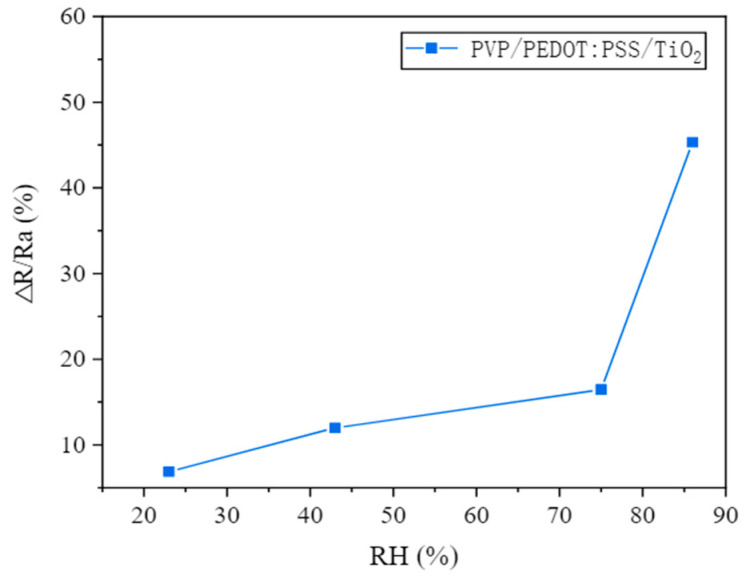
The influence of relative humidity, in the range of 11–86%, on the gas sensitivity response of the PVP/PEDOT:PSS/TiO_2_ sensors [50].

**Figure 8 polymers-16-02514-f008:**
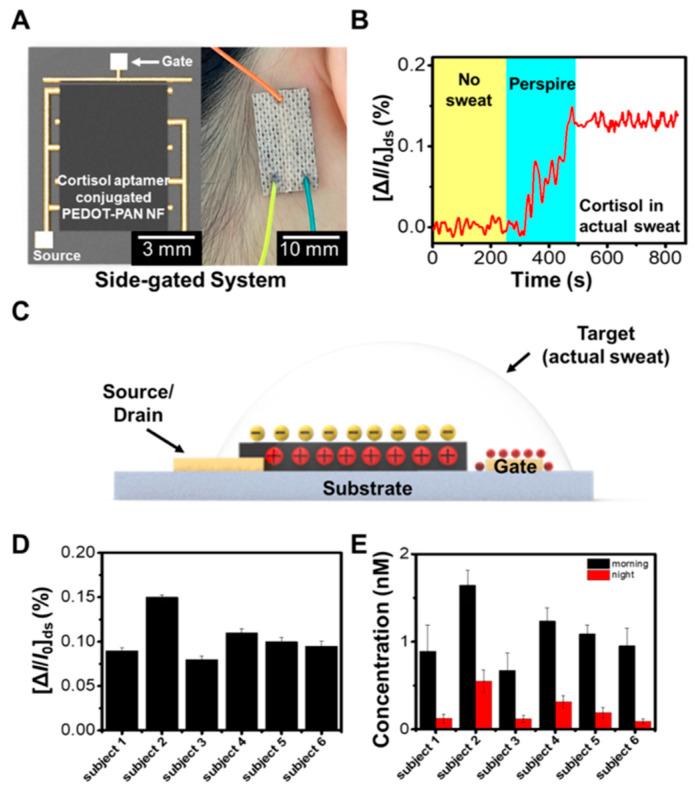
Application of the wearable cortisol aptasensor for detecting cortisol in actual sweat. (**A**) Illustration and optical image of the side-gated system cortisol aptasensor detecting cortisol in actual sweat. (**B**) Real-time monitoring using an actual sweat sample from subject 2. (**C**) Schematic illustration of the side-gated system mechanism. (**D**) Bar graph comparing the intensity change in subjects’ actual sweat samples. (**E**) ELISA results of the subject in the morning (black) and at night (red) after cycling. Reproduced with permission from An et al. in ACS Sensors, published by American Chemical Society, 2022 [60].

**Figure 9 polymers-16-02514-f009:**
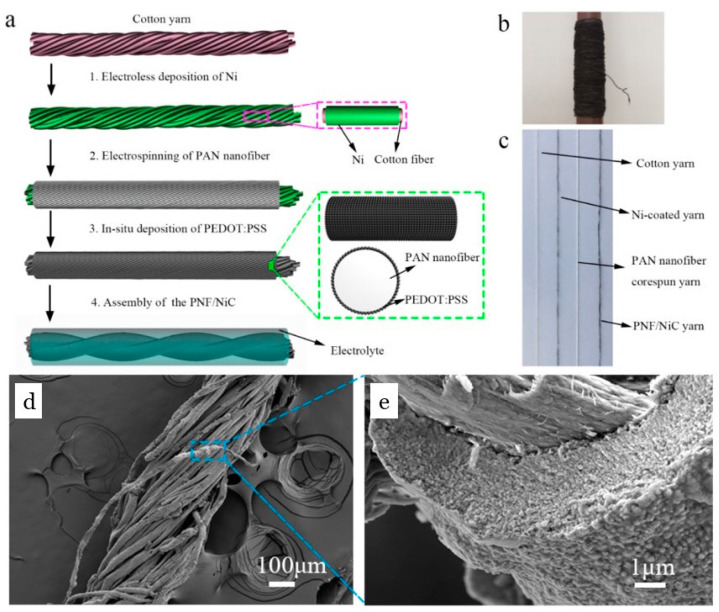
(**a**) Scheme of the fabrication of the PNF/NiC capacitor yarn. (**b**) Photograph of the Ni-coated cotton yarn wound on the spinning cone. (**c**) Photograph of various yarns during processing. (**d**) High-magnification image of polyacrylonitrile nanofiber core-spun yarn and (**e**) its cross-section [78].

**Table 1 polymers-16-02514-t001:** In the table are the main advantages of electrospinning for PEDOT-based nanofibers over other additive manufacturing methods such as 3D printing and bioprinting.

Feature/Method	Electrospinning (PEDOT)	Other A.M. Methods
Surface area	High	Moderate to low
Porosity	High	Variable
Flexibility	High	Low to moderate
Conductivity	Enhanced with additives	Lower without additives
Material versatility	High	Limited
Mechanical properties	Improved with additives	Variable
Morphological control	Precise	Less precise
Applications	Broad (wearables, energy, biomedical)	Limited to specific fields

## Data Availability

No new data were created or analyzed in this study. Data sharing is not applicable to this article.

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
