# Peer review of "Advances in the Fabrication, Properties, and Applications of Electrospun PEDOT-Based Conductive Nanofibers"

_polymers, 2024, doi:10.3390/polym16172514_

Round 1

Reviewer 1 Report

Comments and Suggestions for Authors

Slejko et al. summarized recent progress in electrospun conductive nanofibers based on PEDOT. This review paper is divided into several parts in terms of synthesis, characterizations, properties, and applications. The research works are reviewed in an organized and detailed manner. The reviewer supposes that studying nanofibers for low-cost fabrication methods and their properties are critical to realize novel applications and its commercialization. As a result, I propose that this paper could be published in Polymers after addressing the minor concerns listed below.

1.     The quality of the figures should be improved. Figure 1 and Figure 8 are very blurred, which should be replaced by high-resolution figures. In Figure 8, “d” and “e” should not be labeled with white background.

2.     In the introduction part, a timeline of the development of PEDOT-based conductive nanofibers should be provided for better summarizing this field.

3.     In the Conclusions part, the unique advantage of electrospinning for PEDOT-based conductive Nanofibers should be highlighted compared to other fabrication methods and fiber materials, especially in terms of applications. Providing a table for a comprehensive comparison will be helpful for readers to understand the key points.

4.     The section “6.3.2 Energy Storage” should not be categorized under “6.3 Wearable Electronics”. Instead, it should be listed independently as 6.4 section.

5.     In section 2 Fundamentals of Electrospinning, schemes should be provided for readily understanding each hardware part and parameters of electrospinning.

Author Response

Dear Reviewer,

thank you for the valuable analysis of our manuscript and the inputs provided, we have found them very useful to improve the quality of our research. In accordance with your comments, we have decided to add more details and further explanations. Here below our point-by-point response.

1. The quality of the figures should be improved. Figure 1 and Figure 8 are very blurred, which should be replaced by high-resolution figures. In Figure 8, “d” and “e” should not be labeled with white background.

Figures 1 and 8 (now 9) have been replaced with high resolution ones (Figure 1 has been totally reworked by the authors). Regarding the white background of labels in Figure 9, we cannot get rid of it as it covers the old labels from the original image (which are not consistent with the new ones).

2. In the introduction part, a timeline of the development of PEDOT-based conductive nanofibers should be provided for better summarizing this field.

We thank the reviewer for this comment. A paragraph illustrating the evolution of PEDOT nanofibers, from the first vapour polymerizations on top of insulating polymers to the complex core-shell configurations, has been added in the Introduction.

Page 3: "The evolution of PEDOT nanofibers produced by electrospinning has seen significant advancements, particularly in their conductivity and mechanical properties. Initially, conductive PEDOT nanofibers were fabricated through electrospinning of insulating polymers followed by vapor-phase polymerization of PEDOT, achieving high conductivity and resistive heating capabilities. It was only during the 2010s that studies introduced techniques like oxidative polymerization and the blending of PEDOT with other polymers for solution-based electrospinning, further enhancing the structural stability, conductivity, and flexibility of the nanofibers. In 2017, Yu et al. have been able to propose a one-step fabrication of core-shell nanofibers based on PEDOT:PSS and poly(ethylene oxide), while three years later Du et al. developed a core-shell structure with chitosan nanofiber with improved biocompatibility for biomedical research and regenerative medicine. These innovations have positioned PEDOT nanofibers as promising materials for applications in flexible electronics, sensors, and biomedical devices."

3. In the Conclusions part, the unique advantage of electrospinning for PEDOT-based conductive Nanofibers should be highlighted compared to other fabrication methods and fiber materials, especially in terms of applications. Providing a table for a comprehensive comparison will be helpful for readers to understand the key points.

We added more considerations in the Conclusions, as well as in Section 3 "Synthesis of PEDOT-based nanofibers via electrospinning". The comparison with conventional production methods should clearly indicate the main advantages of ES, also thanks to Table 1 in which we discuss each of the main characteristic parameters in the fabrication process.

Page 6: "Electrospinning offers a unique advantage for fabricating PEDOT-based conductive nanofibers compared to other fabrication methods and fiber materials, particularly in terms of applications. This technique allows for the creation of nanofibers with high surface area, porosity, and flexibility, which are crucial for various advanced applications such as wearable electronics, energy storage, and biomedical devices. The ability to control fiber morphology and composition through electrospinning enhances the performance of PEDOT-based nanofibers, making them superior to other methods and materials. The specific advantages of electrospinning for PEDOT-based conductive nanofibers and provide in the following. 

High surface area and porosity: electrospinning produces nanofibers with a high surface area to volume ratio, which is valuable for applications requiring high reactivity and interaction with the environment, such as sensors and catalysts.

Flexibility and comfort: the nanofibers mats produced are highly flexible, making them suitable for wearable electronics where comfort and adaptability to body movements are essential.

Enhanced conductivity: by incorporating materials like reduced graphene oxide (rGO), the conductivity of PEDOT-based nanofibers can be significantly enhanced, which is crucial for electronic applications.

Versatility in material composition: electrospinning allows for the incorporation of various materials, such as nanocarbons, to enhance the mechanical and electrical properties of the fibers, making them suitable for a wide range of applications from photovoltaics to biomedical devices.

Improved mechanical properties: the process can improve the mechanical properties of the fibers, such as tensile strength, which is important for structural applications.

Controlled morphology: electrospinning allows for precise control over fiber diameter and morphology, which can be tailored to specific application needs, such as creating coaxial structures for electronic devices."

Page 25: "While electrospinning offers significant advantages for PEDOT-based conductive nanofibers, it also presents challenges such as the need for precise control over processing parameters to achieve desired fiber characteristics. Additionally, the inherent low molecular weight and rigidity of conductive polymers like PEDOT can complicate the electrospinning process, often requiring the use of carrier polymers or coaxial spinning techniques to achieve stable fiber formation. Despite these challenges, the versatility and adaptability of electrospinning make it a promising method for advancing the development of high-performance conductive nanofibers across various applications."

4. The section “6.3.2 Energy Storage” should not be categorized under “6.3 Wearable Electronics”. Instead, it should be listed independently as 6.4 section.

We agree with the reviewer and have corrected the section's title accordingly.

5. In section 2 Fundamentals of Electrospinning, schemes should be provided for readily understanding each hardware part and parameters of electrospinning.

We thank the reviewer for this comment: a new scheme (namely Figure 2) has been added with a complete overview of the equipment hardware and parameters' influence to help the reader following the description on the working principles of electrospinning in Section 2.

Page 5: "A conventional solution electrospinning setup, as illustrated in Figure 2, comprises several essential components, including a high-voltage power source, a syringe pump, a syringe loaded with the polymeric solution, a metal spinneret, usually a blunt-cut needle,  connected to the syringe, and a fiber collector, typically a metal plate connected to a grounded electrode. The syringe pump regulates the flow rate of the polymer solution. The high-voltage power supply, in conjunction with the spinneret that directs the charged polymer solution into a high electric field, and ultimately to the collector provides the force necessary to elongate the charged polymer solution into fibrous structures."

Reviewer 2 Report

Comments and Suggestions for Authors

Dear Authors,

I have completed the review of your manuscript titled "Advances in the Fabrication, Properties, and Applications of Electrospun PEDOT-based conductive Nanofibers". I would like to recommend some revisions before further consideration.

1) Please provide more details on the mechanisms behind the enhanced EMI shielding effectiveness observed in the PEDOT-based nanofiber composites.

2) In the context of supercapacitors, how do the PEDOT nanofibers' porous structure enhance ion diffusion and storage capacity compared to traditional materials?

3) What strategies were employed to address the challenges related to scalability and reproducibility of the electrospinning process in producing PEDOT-based nanofibers at a larger scale?

4) Could you elaborate on the green and sustainable electrospinning processes that were considered to minimize the environmental impact of large-scale production? Has a life cycle assessment been conducted to evaluate the environmental impact of PEDOT-based nanofiber production and usage? How do the sustainability aspects of this technology compare to conventional materials?

5) What specific advancements or breakthroughs do you believe this research on PEDOT-based nanofibers can contribute to fields such as healthcare, environmental monitoring, and flexible electronics?

6) Can you elaborate on how the incorporation of composite materials improved the mechanical strength and flexibility of the PEDOT nanofibers? Were there any trade-offs in electrical performance?

7) How was the long-term stability of PEDOT-based nanofibers evaluated, especially in harsh environmental conditions such as exposure to moisture, temperature variations, and UV radiation?

8) Please provide more details on the implementation of real-time monitoring systems during the electrospinning process to ensure consistency in fiber properties. How did this affect the reproducibility of the nanofibers?

Author Response

Dear Reviewer,

thank you for the valuable analysis of our manuscript and the inputs provided, we have found them very useful to improve the quality of our research. In accordance with your comments, we have decided to add more details and further explanations, especially in Section 7. Here below our point-by-point response.

1. Please provide more details on the mechanisms behind the enhanced EMI shielding effectiveness observed in the PEDOT-based nanofiber composites.

We have added the following text in Section 6.3 "Wearable Electronics" to better explain the EMI shielding mechanism.

Page 20: "The EMI shielding is based on reflection and absorption loss. The former is the weakening of the electromagnetic wave when passing through materials with mismatching dielectric constants. The latter is the weakening when propagating inside a material. The reflection loss is enhanced in nanofiber based materials due to the large surface to volume ratio and can be further improved with coaxial or multiwire fiber geometries. Absorption loss can be achieved if the thickness of the fabric is sufficient and can be improved by adding nanofillers with magnetic properties to the fibers."

2. In the context of supercapacitors, how do the PEDOT nanofibers' porous structure enhance ion diffusion and storage capacity compared to traditional materials?

We agree with the reviewer and thank her/him for the comment. The explanation for the enhanced performance is related to the tunability of the free space within the nonwoven material. In fact, by tuning the electrospinning parameters, is it possible to obtain thinner fibers and a lower density of fibers per unit area, therefore improving the diffusion rate of ions within the material.

Page 21: "This enhancement can be explained in terms of hierarchical substructure of the yarns and mats, which can be easily tuned to allow optimized electronic transportation and ion diffusion from the interior of the material to the surface by shortening the ion diffusion distance. The large surface to volume ratio and the easy tunability of the fiber structure are indeed intrinsic advantages of the electrospinning technique. On the other side, efficient electron charge transfer is enabled by the high conductivity of the PEDOT polymer."

3. What strategies were employed to address the challenges related to scalability and reproducibility of the electrospinning process in producing PEDOT-based nanofibers at a larger scale?

Extensive considerations about the scalability and reproducibility have been added to Section 7, as follows.

Page 23: "Scalability is one of the primary challenges of the electrospinning process, which is often limited by the need for precise control over fiber morphology and uniformity. This issue can be addressed by optimizing the electrospinning parameters and developing advanced fabrication techniques that ensure consistent fiber production at a larger scale. Recent strategies include the use of multi-needle systems, auxiliary electrodes, and controlled environmental conditions, each contributing to overcoming specific limitations of traditional electrospinning, focusing on enhancing production capacity and ensuring consistent fiber quality. The use of multiple electrospinning needles operating in parallel is a prominent strategy to increase production capacity. This approach, however, introduces challenges such as uneven electric field distribution due to Coulombic repulsion between charged jets and needles. To mitigate this, auxiliary electrodes have been employed to uniformize the electric field, thereby improving the deposition area and fiber morphology. Implementing parallel lateral plates as auxiliary electrodes has been shown to effectively address the uneven electric field distribution in multi-needle systems. This setup helps achieve a uniform electric field, which is crucial for consistent fiber production. The use of finite element simulations has further validated the impact of these electrodes on the electric field intensity, leading to high-quality polymer nanofibers."

Page 23: "Reproducibility in electrospinning is significantly influenced by environmental conditions, particularly relative humidity (RH). A miniaturized electrospinning setup, designed to fit within a biological safety cabinet, allows precise control of RH, enhancing the reproducibility of fiber production. This setup demonstrated that controlling RH can significantly affect fiber diameter, thus ensuring consistent quality across production batches. While these optimization strategies have advanced the scalability and reproducibility of electrospinning, challenges remain, particularly in balancing high throughput with fiber quality. Future research may focus on integrating these advanced electrospinning configurations with automated systems to further enhance production efficiency and consistency."

4. Could you elaborate on the green and sustainable electrospinning processes that were considered to minimize the environmental impact of large-scale production? Has a life cycle assessment been conducted to evaluate the environmental impact of PEDOT-based nanofiber production and usage? How do the sustainability aspects of this technology compare to conventional materials?

Thank you for your comment. We have added clarification regarding the low impact of electrospinning, mainly considering its precise deposition and reduced waste generation. Unfortunately, LCA studies have not been conducted yet on specific materials, but we are sure it will be addressed in the next future.

Page 24: "In fact, this fabrication technique, which is a form of additive manufacturing, offers several environmental benefits. Firstly, electrospinning is highly efficient, allowing for precise control over material usage, thereby minimizing waste. The process typically operates at ambient temperatures, especially when carried out in a solution-based configuration, reducing energy consumption compared to traditional manufacturing methods. Additionally, the ability to utilize biodegradable or recyclable polymers in electrospinning further enhances its environmental credentials. Furthermore, traditional fiber production methods like Chemical Vapor Deposition (CVD), melt blowing, or solution blow spinning often require large amounts of solvents or other chemicals, which can lead to hazardous waste and environmental pollution. Electrospinning can use greener solvents or even solvent-free approaches, thereby minimizing chemical waste and the environmental impact. However, while the potential for sustainability is clear, comprehensive Life Cycle Assessment (LCA) studies are still needed to fully quantify the environmental impact of large-scale electrospinning and to identify areas for further improvement."

5. What specific advancements or breakthroughs do you believe this research on PEDOT-based nanofibers can contribute to fields such as healthcare, environmental monitoring, and flexible electronics?

The development of flexible, stretchable electronic skins that mimic natural skin is crucial for applications like healthcare monitoring and robotics, but current wearable sensors often lack comfort and face issues with signal interference from different stimuli. We believe PEDOT-based nanofibers could represent a valuable asset in the realization of those innovative applications.

Page 24: "Current wearable sensors often lack comfort and face issues with signal interference from different stimuli. Functional sensors that address these challenges are essential for providing reliable interaction between humans and their environment. PEDOT-based nonwovens are solid candidates for the development, for example, of flexible, stretchable electronic skins that mimic natural skin for integration in smart clothing."

6. Can you elaborate on how the incorporation of composite materials improved the mechanical strength and flexibility of the PEDOT nanofibers? Were there any trade-offs in electrical performance?

While current research is investigating various combinations of PEDOT and other polymers in the presence of additives or solvents, a focus on composites is still in its early stage. Nevertheless, the state-of-the-art indicates interesting effects on the structure of PEDOT due to, for example, DMSO and EG, enhancing its mechanical and electrical properties. By refining the electrospinning process, more complex composite structures will be available to researcher, further improving the characteristics of the final material. 

Page 24: "For example, stretchable conducting polymeric films were fabricated by blending PEDOT:PSS with poly(vinyl alcohol) in polar organic solvents like DMSO and EG, resulting in enhanced electrical and mechanical properties. The study found that DMSO and EG induce crystalline domains and structural changes in PEDOT, significantly boosting electrical performance and stretchability. It is reasonable to expect a similar behavior in nonwoven materials, too."

7. How was the long-term stability of PEDOT-based nanofibers evaluated, especially in harsh environmental conditions such as exposure to moisture, temperature variations, and UV radiation?

We thank the reviewer for the comment. We clarified our statement by adding the following considerations.

Page 24: "Stability, particularly in harsh environmental conditions, is essential for the practical application of these nanofibers. To the best of our knowledge, no specific study has been yet published on the stability of PEDOT-based nanofibers. Nevertheless, few recently published studies addressed the stability of PEDOT-based thin films investigating the aging/degradation mechanism and possible enhancement methods. Such improvements could be obtained by surface modifications, stabilizing agents  or by coatings that protect the fibers from degradation due to moisture, temperature, and UV exposure."

8. Please provide more details on the implementation of real-time monitoring systems during the electrospinning process to ensure consistency in fiber properties. How did this affect the reproducibility of the nanofibers?

Commercial solutions as well as research-level ones for the real-time monitoring of the material quality are now discussed in Section 7, as follows.

Page 25: "Real-time monitoring of nanofiber quality is conducted by means of interferometry: in industrial roll-to-roll setups, fiber thickness can be estimated with an optical interferometer, which provide feedback to the flow rate controller of the electrospinning instrument. Furthermore, a novel optical system integrated through a coaxial needle has been recently proposed to monitor the process, aiming to enhance the reproducibility and quality of nanofibers. The system uses an optical fiber to pass light through the polymeric solution and couple it to the electrospun fiber. Experimental tests confirmed that the system effectively tracks changes in electrospun fiber's waveguide properties and nanofiber diameter, offering a promising solution for industrial-scale electrospinning using multi- or moving needles."

Round 2

Reviewer 2 Report

Comments and Suggestions for Authors

Dear Authors,

I would like to express my appreciation for taking my feedback into account and revising your work. Based on the improvements you have made, I am pleased to recommend the publication of your article in its current form.